# Ketogenic Diet and Cardiac Substrate Metabolism

**DOI:** 10.3390/nu14071322

**Published:** 2022-03-22

**Authors:** Thien Vinh Luong, Caroline Bruun Abild, Maj Bangshaab, Lars Christian Gormsen, Esben Søndergaard

**Affiliations:** 1Department of Nuclear Medicine & PET-Centre, Aarhus University Hospital, Palle Juul-Jensens Blvd. 165, 8200 Aarhus, Denmark; thiluo@clin.au.dk (T.V.L.); lars.christian.gormsen@clin.au.dk (L.C.G.); 2Steno Diabetes Center Aarhus, Aarhus University Hospital, Hedeager 3, 8200 Aarhus, Denmark; caroline.christiansen@rm.dk (C.B.A.); bangshaab@clin.au.dk (M.B.); 3Department of Clinical Medicine, Aarhus University, Palle Juul-Jensens Blvd. 82, 8200 Aarhus, Denmark

**Keywords:** ketogenic diet, ketone bodies, heart, metabolism, heart failure, diabetic cardiomyopathy

## Abstract

The ketogenic diet (KD) entails a high intake of fat, moderate intake of protein, and a very limited intake of carbohydrates. Ketogenic dieting has been proposed as an effective intervention for type 2 diabetes and obesity since glycemic control is improved and sustained weight loss can be achieved. Interestingly, hyperketonemia is also associated with beneficial cardiovascular effects, possibly caused by improved cardiac energetics and reduced oxygen use. Therefore, the KD has the potential to both treat and prevent cardiovascular disease. However, the KD has some adverse effects that could counteract the beneficial cardiovascular properties. Of these, hyperlipidemia with elevation of triglycerides and LDL cholesterol levels are the most important. In addition, poor diet adherence and lack of knowledge regarding long-term effects may also reduce the broader applicability of the KD. The objective of this narrative review is to provide insights into the KD and its effects on myocardial ketone body utilization and, consequently, cardiovascular health.

## 1. Introduction

The ketogenic diet (KD) is a carbohydrate-reduced diet that results in a substantial increase in the level of circulating ketone bodies. The KD has for decades been used to treat drug-resistant epilepsy in children [1] and it is also used as an adjuvant treatment to obtain seizure control in adults with epilepsy [2]. The diet has recently attracted renewed interest due to several promising beneficial effects on various diseases and organs [3]. Thus, the KD has been suggested to have beneficial effects in the treatment of Alzheimer disease [4] and some cancers [5,6] and has been proven efficient as a means to achieve significant weight loss [7]. Even more interest in the KD was sparked by the conclusion of the EMPA-REG study, which showed that sodium-glucose cotransporter-2 (SGLT2) inhibitors reduce cardiovascular morbidity and mortality markedly [8]. It was hypothesized that the cardiovascular benefit was mediated by the observed increase in circulating ketone bodies [9], since ketone bodies are an energy-efficient and oxygen-sparing fuel for the heart. This hypothesis is supported by recent studies by our group showing that acute hyperketonemia by exogenous infusion of ketone salts has potentially beneficial hemodynamic effects in both healthy subjects [10] and in patients with heart failure [11]. However, SGLT-2 inhibition only elevates circulating levels of ketone bodies moderately (≈0.6 mM) [12], and the KD may, therefore, serve as a more attractive manner in which to achieve a substantially higher degree of ketosis. The scope of this narrative review will, therefore, be to provide insights into the metabolism of ketone bodies, how ketone bodies are metabolized by the heart, and finally the therapeutic potential of the KD in selected groups of patients.

## 2. Ketone Bodies

Ketone bodies are produced in the liver with fatty acids and ketogenic amino acids as substrates. Ketogenesis is stimulated by the delivery of fatty acids to the liver and by glucagon released from the pancreas, whereas insulin is a strong inhibitor [13]. Three different ketone bodies are synthesized in-vivo: 3-hydroxybutyrate (3-OHB), acetoacetate, and acetone that is produced by spontaneous decarboxylation of acetoacetate. The 3-OHB constitutes the majority (~80%) of total circulating ketone body concentration [14]. Ketone bodies are used as a source of energy primarily in the heart and brain when carbohydrate availability is limited, such as during fasting or prolonged exercise [15,16]. In the fed state, plasma total ketone body concentration is approximately 0.1 mM. The concentration of ketone bodies gradually increases with prolonged fasting to 1 mM after 24 h and further increases until a plateau of 5–7 mM is reached after several weeks of fasting [17]. In untreated individuals with insulin-deficient diabetes, plasma ketone body levels may increase to more than 20 mM, causing massive metabolic disturbances [13,14]. In the postabsorptive state, ketone bodies supply 5% of total energy consumption, but this may increase to 10–18% during exercise or fasting [18].

Previously, ketone bodies were primarily considered harmful due to their role in the pathogenesis of diabetic and alcoholic ketoacidosis. However, it has become increasingly clear that ketone bodies also have several beneficial effects. First, ketone bodies may provide otherwise fuel-starved organs and tissues with an alternative source of energy during conditions of glucose deprivation, sparing the limited glucose stores in the body [17]. This is of particular importance in the brain, since the quantitatively most important circulating source of energy (fatty acids) do not cross the blood–brain barrier. Second, ketone bodies require less oxygen than fatty acids to generate a similar amount of ATP [14], which may protect periodically oxygen-deprived organs such as the ischemic heart with an oxidatively *efficient* source of energy. In this context, it is tempting to speculate that the increase in ketone bodies observed in patients with heart failure constitutes a compensatory mechanism to counteract hypoxia [19].

## 3. Ketone Bodies and the Heart

### 3.1. The Healthy Heart

The heart is often described as a metabolic omnivore since it readily utilizes all energy substrates as fuel and easily flexes between different compositions of energy substrates depending on energy demand and substrate availability (Figure 1). In the healthy, postabsorptive heart, fatty acids account for around 70% of cardiac energy consumption [19,20], whereas glucose and ketone bodies have been estimated to deliver less than 10% of the total energy demand [21]. Even this modest contribution from glucose and ketone bodies may be an overestimation, as evidenced by a recent elegant metabolomics’ study, in which glucose uptake was barely measurable and was estimated to account for less than 1% of total energy demand [22]. The remaining myocardial energy need is derived from oxidation of lactate and branched-chain amino acids [21,23]. Glucose, fatty acids, ketone bodies, and branched-chain amino acids are, thus, all sources of acetyl-CoA feeding into the Krebs cycle [14] ultimately generating ATP. Therefore, oxidation of a given substrate primarily occurs in proportion to the delivery of the substrate into the cardiomyocyte at the expense of other substrates [20]. In the postprandial state, carbohydrate consumption stimulates insulin secretion, resulting in increased myocardial glucose oxidation at the expense of fatty acid oxidation. This is mediated through (1) an increased glucose delivery through insulin-mediated glucose uptake in the cardiomyocyte, and (2) insulin-mediated inhibition of adipose tissue lipolysis with reduced fatty acid release and, therefore, also less fatty acid delivery to the heart. This was demonstrated in an elegant study by Ferrannini et al. [24], who used cardiac vein and arterial catherization during a hyperinsulinemic-euglycemic clamp to demonstrate that hyperinsulinemia dramatically increases myocardial uptake and utilization of glucose, lactate, and pyruvate. By contrast, myocardial extraction of circulating fatty acids, glycerol, and 3-OHB was almost completely absent. Interestingly, the shift from fat to carbohydrate metabolism did not change cardiac oxygen consumption, heart rate, blood flow, or cardiac pressure and work parameters. This flexibility ensures a constant supply of cardiac fuel even during rapidly changing hormonal and metabolic conditions. However, the study by Ferrannini et al. did not explore whether glucose is preferentially oxidized during conditions of ample supply of both carbohydrate and lipid fuel sources, including ketone bodies.

We recently addressed this question in a study designed to determine whether hyperketonemia during conditions of hyperinsulinemia and euglycemia alters myocardial glucose oxidation. In a crossover design in healthy subjects, we used positron emission tomography (PET)/computed tomography (CT) to examine the effects of an intravenous ketone salt infusion on cardiac glucose and fatty acid metabolism [10]. Most noteworthy was the observation that an acute increase (~4 h) in ketone body levels to 3.8 mM resulted in a 50% reduction in myocardial glucose uptake despite maximal insulin stimulation and an ample supply of glucose. Thus, ketone bodies appear to be preferred over glucose when both substrates are available [10]. Myocardial fatty acid uptake remained unchanged during the study; however, it should be noted that circulating fatty acid levels were depressed by the antilipolytic effects of the hyperinsulinemic clamp. Of interest, the preferential uptake and oxidation of ketone bodies at the expense of glucose appears to be specific for the heart and brain [25] since hyperketonemia did not affect glucose and fatty acid uptake in other organs (skeletal muscle, kidney, spleen, liver, or subcutaneous or visceral fat) [26].

### 3.2. Heart Failure

Heart failure is characterized by alterations in cardiac substrate metabolism, structural remodeling, and impaired contractility. The heart failure-related changes in myocardial metabolism are thought to be to some extent caused by mitochondrial oxidative dysfunction [23], resulting in a less effective utilization of energy substrates for ATP production. In addition, the failing heart becomes less flexible in shifting between energy substrates depending on their availability. The result is a sometimes paradoxically energy-starved [27] heart, fittingly described by wiser people as “an engine out of fuel” [28]. The failing heart is also characterized by impaired insulin signaling and insulin stimulation of glucose oxidation, resulting in a progressive development of cardiac insulin resistance [23]. As a consequence of the reduced ATP production by glucose oxidation, glycolytic processes are frequently upregulated, particularly evident in ischemic heart failure [29]. Moreover, fatty acid oxidation is decreased in tandem with the increase in glycolysis [21]. In this setting, it is becoming increasingly clear that the failing heart may rely much more on ketone bodies as a source of energy than the healthy heart [23,30]. In fact, it has been shown that ketone bodies account for 16% of total cardiac ATP generation in heart failure patients [22] and that the ketolytic enzymes BDH1 and SCOT are upregulated in the failing heart [31]. It is plausible that this is a metabolic adaptation to compensate for an otherwise ineffective substrate utilization and increased oxygen demand [32].

Ketone bodies may also have more direct effects on cardiac contractile function in individuals with heart failure. This was explored in a study by Nielsen et al., who demonstrated a dose-dependent increase in cardiac output of up to 2 L/min and an 8% increase in left ventricular ejection fraction as a result of an exogenous infusion of ketone bodies [11]. Somewhat disappointingly, however, the ketone body infusion did not improve myocardial external efficiency (MEE), which is impaired in heart failure and usually changes in a characteristic manner during medical or surgical cardiac interventions [33]. This lack of improvement in MEE implies that at least an acute elevation of ketone bodies may not have the anticipated oxygen-sparing effect. However, in that study MEE was measured during a hyperinsulinemic-euglycemic clamp and a sizable fraction of ATP production could, therefore, have been derived from glucose oxidation. It is possible that MEE could have been increased by an exogenous ketone body infusion performed in the postabsorptive state, where myocardial energy consumption is overwhelmingly served by relatively oxygen-inefficient fatty acids.

### 3.3. The Diabetic Heart Disease

The impact of ketone bodies on cardiac substrate metabolism has not been thoroughly studied in patients with diabetes. Diabetes is characterized by increased levels of circulating fatty acids and hyperglycemia; but, despite the increased delivery of glucose to the coronary circulation, the diabetic heart almost exclusively oxidizes fatty acids [34]. Much as in heart failure, the diabetic heart is metabolically inflexible and energetically inefficient, increasing the risk of myocardial injury during bouts of ischemia [14].

Individuals with diabetes may develop diabetic cardiomyopathy that does not necessarily resemble the typical cardiac pathology seen in patients with ischemic heart failure. Diabetic cardiomyopathy is characterized by early diastolic defects, interstitial fibrosis, and left ventricular hypertrophy followed by systolic dysfunction, which is then recognized as clinical heart failure. Diabetic cardiomyopathy can be observed in individuals with T2D even in the absence of the traditional risk factors for heart failure (hypertension, dyslipidemia, obesity, and coronary artery disease) [35,36] and carries a 2.5-fold increased risk of heart failure [37]. Independent risk factors for developing diabetic cardiomyopathy include insulin resistance, hyperinsulinemia, and hyperglycemia. It was hypothesized that diabetic cardiomyopathy is caused by inappropriate activation of the renin–angiotensin–aldosterone system, oxidative stress, lipotoxicity, inflammation, and dysfunctional immune modulation [38].

Overt diabetes is associated with markedly reduced insulin-mediated suppression of lipolysis (adipose tissue insulin resistance) [39], resulting in release of excess fatty acids that are converted to ketone bodies by the liver. Further adding to circulating ketone levels in diabetes, there are also indications that hepatic insulin resistance may result in increased hepatic ketogenesis [13]. Both individuals with and without type 2 diabetes take up ketone bodies in the heart at the expense of glucose, lactate, and pyruvate, as shown by Mizuno et al. using cardiac catherization of the coronary sinus and the aortic root [40]. It is conceivable that increased ketone body oxidation could serve as an oxygen-sparing adaption in the diabetic heart, which otherwise primarily oxidizes fatty acids. This shift in myocardial substrate oxidation is also one of the hypothesized benefits of SGLT-2 inhibitor treatment, where ketone levels increase to ~0.6 mM in individuals with type 2 diabetes [12]. Such a modest ketonemia does not closely mimic the levels of circulating ketone bodies achieved by infusions of ketone body salts (3–4 mM); it is, therefore, not surprising that 4 weeks of treatment with the SGLT2 inhibitor empagliflozin (resulting in ketone body levels of ~0.1–0.2 mM) does not affect PET-measured cardiac substrate metabolism or myocardial efficiency [41].

## 4. The Ketogenic Diet and the Heart

Whereas pharmacological intervention may increase circulating ketone bodies to a modest degree, dietary interventions may potentially result in vastly more pronounced and sustained hyperketonemia. Of these, the KD has been investigated most frequently. The KD entails a strict reduction in carbohydrate intake to reduce insulin secretion and, by extension, insulin-mediated inhibition of ketogenesis. The diet belongs to a group of low-carbohydrate diets (LCD), which usually provide 50–150 g of carbohydrates, equivalent to 10–30% of the daily caloric intake. Diets with higher amounts of carbohydrate intake do not impact ketone body levels and will not be discussed further [42]. One of the main problems evaluating the effect of LCD has turned out to be a lack of diet standardization. Numerous diets are within the definitions of LCD but with substantial differences in fat and protein content as well as total caloric content. The classic KD contains much fat (55 to 90% of total calories), moderate amounts of protein (15–35% of total calories), and a very low carbohydrate content (5–10% of total calories) [43,44,45]. Whereas most dietary fats from standard foods are long-chain triglycerides (LCT), medium-chained triglycerides (MCT) primarily containing octanoic and decanoic acid yield more ketone bodies per kilocalorie of energy compared to LCT [46]. In addition, fatty acids from MCT are absorbed directly and delivered to the liver through the portal system. However, MCT’s can only be ingested as supplements in the form of oils and can, therefore, not be labeled a diet in itself. In addition, the ketogenic potential of ingesting MCT is often attenuated by the accompanying ingestion of carbohydrates and protein. In the modified Atkins diet (MAD), around 50–65% of the calories are derived from standard fat sources [47] with a larger protein intake compared to the KD. The MAD is more palatable and less restrictive than the KD, thus increasing compliance in patients with behavioral problems and children treated by diets for epilepsy [46]. However, the greater protein content of the MAD reduces the ketogenic effect, since amino acids from the ingested protein stimulate insulin secretion and thereby inhibit ketogenesis [48]. Furthermore, the ingested amino acids are also converted into glucose through gluconeogenesis, which can lead to higher glucose levels and stimulation of insulin secretion. Since no clear consensus exists regarding the specific quantity and quality of each macronutrient in LCD, it is difficult to compare the different studies and determine to what extent these different diets impact substrate metabolism. Overall, the KD is the most carbohydrate restrictive diet, providing less than 50 g and sometimes even less than 20 g of carbohydrates per day. This results in a greater increase in ketone bodies than what is observed during other LCDs [42,43]. In fact, 3-beta-hydroxy-butyrate (3-OHB) may increase to 3.2 mM after only 4 days on a KD [49] and to 5.2 mM for children on a 12-month classical, strict KD [50].

There are several potential metabolic benefits associated with the KD. The diet induces weight loss at an average of 5% of body weight after 6 months [7], which is slightly more than what can be achieved by low-fat and medium-fat diets [51,52]. However, this may, at least in part, be explained by a greater reduction in fat-free mass, through a loss of water bound in glycogen stores. It has been proposed that the KD reduces caloric intake due to an increased satiety effect of proteins compared to carbohydrates [53] and a reduction in appetite induced by ketosis reflected by lower ghrelin concentrations [54]. Moreover, the KD contains potential benefits for individuals with type 2 diabetes since it decreases fasting blood glucose levels [55], HbA1c [6,56], and glycemic variability [57]. It has also been shown to reduce the need for antidiabetic medications in individuals with type 2 diabetes [58].

Some important potential cardiovascular side effects of the KD should be acknowledged. First, a significant side effect of the KD is a substantial rise in triglycerides and low-density lipoprotein (LDL) cholesterol in the first months of a KD [59]. However, the increase in LDL levels is accompanied by an increase in size and volume of LDL cholesterol particles, which may decrease the atherogenicity of the LDL particles [60]. However, since there is a clear causal link between LDL levels and the development of atherosclerosis [61], there is an important concern when balancing the pros and cons of the KD, especially in individuals with cardiovascular disease. Second, a transient development of endothelial dysfunction has been observed 5 days after initiation of a KD, which could further distress an already strained atherosclerotic vascular system [62]. Third, the high levels of circulating free fatty acids could lead to lipotoxicity and promote myocardial insulin resistance through an inhibition of glucose oxidation and insulin signaling in the failing heart [23]. However, myocardial insulin resistance does not appear to have a substantial clinical impact in individuals with ischemic heart failure [63]. Fourth, short chain fatty acids (SCFA), mainly acetate and butyrate, are the product of fermentation in the gut of indigestible food such as dietary fiber. These fatty acids appear to improve insulin sensitivity and overall metabolic health through actions on lipid metabolism and glucose homeostasis [64]. However, the intake of dietary fiber is very limited during a KD, and 1 month of a KD reduces SCFA production [65].

One of the main problems with the KD is adherence to the diet. Short-term side effects, termed the “keto flu”, include constipation, headache, halitosis, muscle cramps, diarrhea, vomiting, and general weakness, which may limit adherence during the first couple of weeks of the diet [66]. Long-term adherence of the KD is difficult to achieve since compliance is difficult to maintain for more than a few months. A strict reduction in carbohydrate intake has proven exceedingly difficult, even for adults treated with a KD for intractable epilepsy. Long-term compliance rates have been reported as low as 45% due to an intolerance caused by the side effects, psychosocial factors, or the restrictiveness of the diet [1].

## 5. Future Directions and Perspective

The KD represents an attractive manner in which to achieve ketosis by promoting endogenous production of ketone bodies. The proposed beneficial effects of hyperketonemia on myocardial metabolism indicate that the KD may be an interesting option for prevention and treatment of heart failure and ischemic heart disease. However, human studies on the cardiac effects of a KD are limited and further studies are needed. Future studies should evaluate the effect of the KD on cardiac perfusion, work, and substrate metabolism as well as the extent of ketosis necessary to obtain these effects.

Whether the KD can safely be used by patients with cardiovascular diseases for prolonged time periods remains largely unanswered. There are indications that the low-carbohydrate, high-protein diet is associated with a higher mortality, whereas the KD is not [67,68]; however, these studies need to be expanded with strictly controlled dietary regimens in terms of sources of proteins and fat and adherence to the diet. Therefore, large, randomized, controlled trials are needed to determine the effect of KD on cardiovascular outcomes and long-term safety.

## 6. Conclusions

Ketogenic dieting is an intriguing non-pharmacological option for treatment and prevention of cardiovascular disease, especially heart failure. However, the effects of the KD on cardiac substrate metabolism are still largely unknown and more longitudinal studies of myocardial metabolism should be undertaken in relevant patient groups before the diet can be implemented in a clinical setting.

## Figures and Tables

**Figure 1 nutrients-14-01322-f001:**
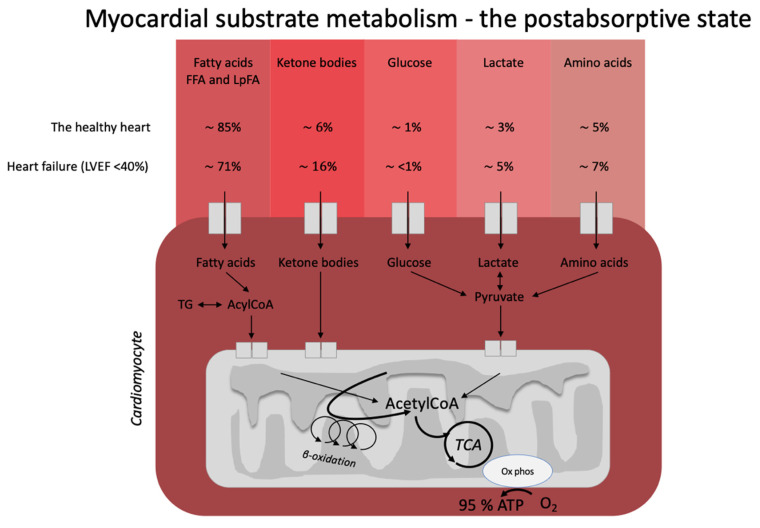
Postabsorptive myocardial substrate metabolism. In the healthy heart, the majority of energy expenditure arises from oxidation of fatty acids from FFA and circulating lipoproteins. In individuals with heart failure, oxidation of ketone bodies (3-hydroxybutyrate, acetoacetate) is upregulated at the expense of fatty acid oxidation. The estimates in the figure were extracted from the recent paper by Murashige et al. [22]. FFA: free fatty acids; LpFA: fatty acids from circulating lipoproteins; LVEF: left ventricle ejection fraction, ATP: adenosine triphosphate, TCA: tricarboxylic acid cycle, AcetylCoA: acetyl coenzyme A, TG: triglycerides.

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
