# Peer review of "Ketogenic Diet and Cardiac Substrate Metabolism"

_nutrients, 2022, doi:10.3390/nu14071322_

Round 1
Reviewer 1 Report
dear colleagues, thank you very much for the paper.
very interesting some references to be read and added:
Dowis K, Banga S. The Potential Health Benefits of the Ketogenic Diet: A Narrative Review. Nutrients. 2021 May 13;13(5):1654. doi: 10.3390/nu13051654. PMID: 34068325; PMCID: PMC8153354.
Di Raimondo D, Buscemi S, Musiari G, Rizzo G, Pirera E, Corleo D, Pinto A, Tuttolomondo A. Ketogenic Diet, Physical Activity, and Hypertension-A Narrative Review. Nutrients. 2021 Jul 27;13(8):2567. doi: 10.3390/nu13082567. PMID: 34444726; PMCID: PMC8398985.
Leone A, De Amicis R, Lessa C, Tagliabue A, Trentani C, Ferraris C, Battezzati A, Veggiotti P, Foppiani A, Ravella S, Bertoli S. Food and Food Products on the Italian Market for Ketogenic Dietary Treatment of Neurological Diseases. Nutrients. 2019 May 17;11(5):1104. doi: 10.3390/nu11051104. PMID: 31108981; PMCID: PMC6566354.
DÄ…bek A, Wojtala M, Pirola L, Balcerczyk A. Modulation of Cellular Biochemistry, Epigenetics and Metabolomics by Ketone Bodies. Implications of the Ketogenic Diet in the Physiology of the Organism and Pathological States. Nutrients. 2020 Mar 17;12(3):788. doi: 10.3390/nu12030788. PMID: 32192146; PMCID: PMC7146425.
Neves GS, Lunardi MS, Lin K, Rieger DK, Ribeiro LC, Moreira JD. Ketogenic diet, seizure control, and cardiometabolic risk in adult patients with pharmacoresistant epilepsy: a review. Nutr Rev. 2021 Jul 7;79(8):931-944. doi: 10.1093/nutrit/nuaa112. PMID: 33230563.
Figure 1 needs to be bigger one page maybe and please provide a colored version. It makes it more citable. Acetyl-CoA is produced in the liver when fatty acids beta-oxide in specific metabolic conditions need to be highlighted. Acetoacetate and beta-hydroxybutyrate are the two ketone bodies.
3.3 The diabetic heart better to be The diabetic heart disease
The future directions are not very clear. Please add clear suggestions for RCTs I think as you implied by safety.
Add statement about ethics: this is a review and it's not applicable check mdpi standard form.
Author Response
Response to reviewers
We thank the reviewers for their insightful comments and suggestions for improving the manuscript. All changes made to the manuscript have been highlighted.
Reviewer 1:
- Dear colleagues, thank you very much for the paper. Very interesting some references to be read and added:
Dowis K, Banga S. The Potential Health Benefits of the Ketogenic Diet: A Narrative Review. Nutrients. 2021 May 13;13(5):1654. doi: 10.3390/nu13051654. PMID: 34068325; PMCID: PMC8153354.
Di Raimondo D, Buscemi S, Musiari G, Rizzo G, Pirera E, Corleo D, Pinto A, Tuttolomondo A. Ketogenic Diet, Physical Activity, and Hypertension-A Narrative Review. Nutrients. 2021 Jul 27;13(8):2567. doi: 10.3390/nu13082567. PMID: 34444726; PMCID: PMC8398985.
Leone A, De Amicis R, Lessa C, Tagliabue A, Trentani C, Ferraris C, Battezzati A, Veggiotti P, Foppiani A, Ravella S, Bertoli S. Food and Food Products on the Italian Market for Ketogenic Dietary Treatment of Neurological Diseases. Nutrients. 2019 May 17;11(5):1104. doi: 10.3390/nu11051104. PMID: 31108981; PMCID: PMC6566354.
DÄ…bek A, Wojtala M, Pirola L, Balcerczyk A. Modulation of Cellular Biochemistry, Epigenetics and Metabolomics by Ketone Bodies. Implications of the Ketogenic Diet in the Physiology of the Organism and Pathological States. Nutrients. 2020 Mar 17;12(3):788. doi: 10.3390/nu12030788. PMID: 32192146; PMCID: PMC7146425.
Neves GS, Lunardi MS, Lin K, Rieger DK, Ribeiro LC, Moreira JD. Ketogenic diet, seizure control, and cardiometabolic risk in adult patients with pharmacoresistant epilepsy: a review. Nutr Rev. 2021 Jul 7;79(8):931-944. doi: 10.1093/nutrit/nuaa112. PMID: 33230563.
R1: We thank the reviewer for the suggestions. All of the references have now been added to the manuscript with accompanying text.
- Figure 1 needs to be bigger one page maybe and please provide a colored version. It makes it more citable. Acetyl-CoA is produced in the liver when fatty acids beta-oxide in specific metabolic conditions need to be highlighted. Acetoacetate and beta-hydroxybutyrate are the two ketone bodies.
R2: We thank the reviewer for the comments. Figure 1 has now been changed to a colored version and has been added inside the manuscript as requested by Nutrients, which can be scaled to a bigger size. The original file with the figure will also be attached in a high resolution. The ketone bodies acetoacetate and beta-hydroxybutyrate have now been highlighted in the figure legend.
- 3 The diabetic heart better to be “The diabetic heart disease”
R3: A good suggestion by the reviewer. This has been changed in the manuscript.
- The future directions are not very clear. Please add clear suggestions for RCTs I think as you implied by safety.
R4: We thank the reviewer for the suggestion. We have added the need for RCTs regarding the long-term effects and safety of the ketogenic diet to the manuscript (line 281-283)
- Add statement about ethics: this is a review and it's not applicable check mdpi standard form.
R5: We are grateful for the reminder. An ethics statement has now been added to the manuscript stating that it is not applicable since the article is a review (line 298).
Reviewer 2 Report
Ketogenic diet and cardiac substrate metabolism
This review article discussed the role of ketogenic diet in cardiac substrate metabolism. This manuscript was well-written. Only few questions were considered.
- How about ketogenic diet in insulin resistance (IR)? Insulin modulated ketogenesis. IR has been known as a risk factor for cardiovascular disease. Please provide this information.
- Please describe the impact of ketogenic diet in short chain fatty acid (SCFA). SCFA has well-known benefit for insulin resistance, metabolic disease and cardiovascular disease.
- Is any suggestion about the duration or dose of ketogenic diet for cardiovascular disease prevention in real world?
Author Response
Response to reviewer 2
We thank the reviewers for their insightful comments and suggestions for improving the manuscript. All changes made to the manuscript have been highlighted.
Reviewer 2:
This review article discussed the role of ketogenic diet in cardiac substrate metabolism. This manuscript was well-written. Only few questions were considered.
- How about ketogenic diet in insulin resistance (IR)? Insulin modulated ketogenesis. IR has been known as a risk factor for cardiovascular disease. Please provide this information.
This is an important topic and there is indeed evidence that indicate that the KD may have intriguing effects in persons with insulin resistance/type 2 diabetes. In a recent study non-randomized 2-year follow-up study, KD improved HbA1c, fasting glucose, and fasting insulin [1]. Also, the mean dose of prescribed insulin decreased by 81% and complete remission of diabetes was seen in 6.7%. In general, growing evidence suggest that following a ketogenic diet, therefore a significantly restriction of carbohydrate consumption, might be a great tool to treat people with insulin resistance [2]. We discuss the impact of diabetes on cardiac substrate metabolism in our manuscript. However, we find it outside the scope of the present review to discuss the effects of KD on insulin resistance in general but acknowledge that this could be an important mediator of the potential cardiovascular benefits. We briefly touch upon the impact of cardiac insulin resistance on cardiovascular prognosis (line 250-254)
- Please describe the impact of ketogenic diet in short chain fatty acid (SCFA). SCFA has well-known benefit for insulin resistance, metabolic disease and cardiovascular disease.
As the reviewer has correctly pointed out, a high level of SCFA have been shown to have a beneficial impact on insulin sensitivty and overall metabolic health through effects on lipid metabolism and glucose homeostasis. Unfortunately, it appears that just one month of ketogenic dieting significant lowers the production of SCFA in the gut. We have added this negative effect of the ketogenic diet in the manuscript (line 254-259).
- Is any suggestion about the duration or dose of ketogenic diet for cardiovascular disease prevention in real world?
R3: We thank the reviewer for this important question. However, as mentioned in the manuscript, we cannot really provide a good answer. The differences in diet composition and the limited reporting of ketone body levels during the diet in the previous studies makes it difficult to provide valid suggestions for duration and dose/level of ketosis. Furthermore, studies have been short, observational and without hard cardiovascular outcomes which further reduces the knowledge obtained from the studies. In a study by Hallberg et al., the KD was followed for one year in individuals with type 2 diabetes [3]. Here, the beta-hydroxybutyrate levels during KD was around 0.4-0.6 mM which improved HbA1C and reduced insulin dose as mentioned above. This also had effects on the lipid profile. However, the observed changes could potentially both increase (increase in LDL cholesterol) or decrease (increase in HDL cholesterol and reduction in triglyceride) the risk of cardiovascular disease. Therefore, the long-term effect on cardiovascular events and safety remains to be elucidated. We now acknowledge this knowledge gap in the manuscript (line 281-283).
References
- Athinarayanan, S.J., R.N. Adams, S.J. Hallberg, A.L. McKenzie, N.H. Bhanpuri, W.W. Campbell, J.S. Volek, S.D. Phinney, and J.P. McCarter, Long-Term Effects of a Novel Continuous Remote Care Intervention Including Nutritional Ketosis for the Management of Type 2 Diabetes: A 2-Year Non-randomized Clinical Trial. Front Endocrinol (Lausanne), 2019. 10: p. 348.
- Dowis, K. and S. Banga, The Potential Health Benefits of the Ketogenic Diet: A Narrative Review. Nutrients, 2021. 13(5).
- Hallberg, S.J., A.L. McKenzie, P.T. Williams, N.H. Bhanpuri, A.L. Peters, W.W. Campbell, T.L. Hazbun, B.M. Volk, J.P. McCarter, S.D. Phinney, and J.S. Volek, Effectiveness and Safety of a Novel Care Model for the Management of Type 2 Diabetes at 1 Year: An Open-Label, Non-Randomized, Controlled Study. Diabetes Ther, 2018. 9(2): p. 583-612.